# Recent Advances on PEO-PCL Block and Graft Copolymers as Nanocarriers for Drug Delivery Applications

**DOI:** 10.3390/ma16062298

**Published:** 2023-03-13

**Authors:** Maria Chountoulesi, Dimitrios Selianitis, Stergios Pispas, Natassa Pippa

**Affiliations:** 1Section of Pharmaceutical Technology, Department of Pharmacy, School of Health Sciences, National and Kapodistrian University of Athens, Panepistimioupolis Zografou, 15771 Athens, Greece; 2Theoretical and Physical Chemistry Institute, National Hellenic Research Foundation, 48 Vassileos Constantinou Avenue, 11635 Athens, Greece

**Keywords:** poly(ethylene oxide)-poly(ε-caprolactone) (PEO-PCL), block copolymer, graft copolymer, drug delivery nanosystems, micelles, hybrid liposomes, polymeric nanoparticles, self-assemble

## Abstract

Poly(ethylene oxide)-poly(ε-caprolactone) (PEO-PCL) is a family of block (or graft) copolymers with several biomedical applications. These types of copolymers are well-known for their good biocompatibility and biodegradability properties, being ideal for biomedical applications and for the formation of a variety of nanosystems intended for controlled drug release. The aim of this review is to present the applications and the properties of different nanocarriers derived from PEO-PCL block and graft copolymers. Micelles, polymeric nanoparticles, drug conjugates, nanocapsules, and hybrid polymer-lipid nanoparticles, such as hybrid liposomes, are the main categories of PEO-PCL based nanocarriers loaded with different active ingredients. The advantages and the limitations in preclinical studies are also discussed in depth. PEO-PCL based nanocarriers could be the next generation of delivery systems with fast clinical translation. Finally, current challenges and future perspectives of the PEO-PCL based nanocarriers are highlighted.

## 1. Introduction

Polymer materials are widely used in the field of pharmaceutical technology for the design and development of different carriers for the delivery and targeting of active pharmaceutical ingredients (APIs). In the recent years, polymer materials have been of paramount importance in nanodelivery as drug delivery platforms of anticancer APIs, proteins, antioxidants, antigens for vaccine delivery, as well as nanoreactors and artificial organelles [1,2,3,4,5].

Poly(ethylene oxide)-poly(ε-caprolactone) (PEO-PCL) is a family of block (or graft) copolymers with several biomedical applications. These types of copolymers are well-known for good biocompatibility and biodegradability properties, being ideal for biomedical applications and for the formation of a variety of nanosystems intended for controlled drug release [6,7,8,9,10,11]. They exhibit also great potential in tissue engineering and medicinal chemistry, too. Several types of particles can be formed by using PEO-PCL block or graft copolymers, ranging from nano- to micro-structures. In this review, we will focus only on the nanocarriers formed by the utilization of PEO-PCL copolymers. In Figure 1, the structure and the chemical architecture of PEO-PCL block or graft polymers as well as the different structures than can be obtained by self-assembly and formulation techniques are illustrated. PEO is a hydrophilic, non-ionic block with low immunogenicity and high blood compatibility. It is already used to modify the surface of liposomal marketed nanomedicines. Due to its hydrophilicity, the interactions with plasma proteins are limited, forming “stealth” nanoparticles. On the other hand, PCL is a hydrophobic, biocompatible, and biodegradable polymer with semi-crystalline properties [6,7,8,9,10,11]. PCL based formulations have been already used for tissue engineering applications, too [12].

The PEO-b-PCL block copolymers are used in drug delivery because they can ameliorate the absorption, distribution, metabolism, excretion, and toxicology (ADMET) profile of the encapsulated active pharmaceutical ingredients (APIs). They can be used for active targeting to improve drug cellular internalization or for passive targeting (enhanced permeability and retention; EPR effect). Last but not least, the degradability of PCL block in acidic pH, gives the opportunity to formulation scientists to explore triggered drug release mechanisms. The stealth properties of the nanostructures self-assembled by PEO-PCL copolymers have been reported by in vitro and in vivo studies. The formation of a protein corona is limited due to the hydrophilic PEO block. The last property is very important for extended circulation times in the human body by intravenous (iv) administration. In addition to drug delivery purposes, PEO-PCL have been already used for Imaging and diagnostic purposes. Low and high molecular weight APIs have been encapsulated in PEO-PCL nanocarriers by different methods, i.e., incorporation into the PCL core, conjugation in the PEO polymeric chains, etc. [13].

The aim of this review is to present the applications and the properties of different nanocarriers rooting from PEO-PCL block and graft copolymers. Firstly, the different synthesis protocols of PEO-PCL copolymers will be presented, giving emphasis to their molecular architecture. Micelles, polymeric nanoparticles, polymersomes, drug conjugates, nanocapsules, and hybrid lipid–polymer nanoparticles as liposomes and other lipidic nanoparticles are the main categories of PEO-b-PCL based nanocarriers loaded with different APIs (Figure 1). The advantages and the limitations in preclinical studies will be also discussed in depth. Due to their intriguing and multifaceted properties, PEO-PCL based nanocarriers could be the next generation of delivery systems with fast clinical translation (Figure 2). Finally, current challenges and future perspectives of the PEO-PCL based nanocarriers will be highlighted.

## 2. Synthesis and Characterization of PEO–PCL Block and Graft Copolymers

Amphiphilic copolymers with different macromolecular architectures consisting of poly(ethylene oxide) (PEO) as a hydrophilic component and poly(ε-caprolactone) (PCL) as a hydrophobic component have been reported in many scientific studies. Here, we will report on the most recent studies on these polymers in terms of their synthesis, physicochemical characterization, and potential applications as nanocarriers in the biomedical field. 

Cho et al. synthesized a novel series of amphiphilic poly(ethylene glycol)-block-poly(ε-caprolactone)-block-poly(ethylene glycol) (PEO–b-PCL-b-PEO) triblock copolymers via ring opening polymerization (ROP). The main aim of this research was to investigate the influence of the ε-caprolactone to ethylene glycol ratio (i.e., [CL]/[EO]) on the physical properties of water-in-oil-in-water multiple emulsions encapsulating a model reagent, ascorbic acid-2-glucoside (AA2G). These copolymers were characterized by a plethora of techniques to confirm and evaluate a variety of parameters such as the molecular weights and compositions, thermal behavior and crystal formation, as well as the sizes and morphology of PEO-PCL-PEO polymeric nanoparticles. The alternation of [CL]/[EO] influences the crystalline temperature and spherulite morphology of these copolymers. As the PCL block increases, it was found that the size of aggregates decreases. Moreover, the copolymers with highest [CL]/[EO] enhanced the droplet size reduction of multiple emulsions and the formation of a stable wall structure. These PEO-PCL-PEO triblock copolymers presented a significant elasticity and colloidal stability. Thus, they could be used as a wall material for the entrapment of multiple emulsions applied in transdermal delivery [14]. 

Along the same line, another interesting work was reported by Cho and co-workers. They developed fluorescein isothiocyanate (FITC)-labeled/PEO–PCL–PEO triblock copolymers. Initially, by utilizing ROP polymerization, they were able to prepare the ε-caprolactone (CL) block where monomethoxy poly(ethylene glycol) (mPEG, Mn = 2000 g mol^−1^) was utilized as the macro-initiator. Two FITC-labeled/PEO-PCL-PEO copolymers with different ratio of hydrophobic to hydrophilic components ([CL]/[EO]) were synthesized in order to evaluate the penetration behavior of them in the hairless mouse skin. By using ultraviolet-visible spectroscopy (UV-Vis) two strong absorption peaks at 489 and 455 nm were observed, corresponding to free FITC and H-aggregated FITC moieties, respectively. The copolymer with high [CL/EO] presented a strong H-aggregation of FITC, resulting in the reduction of fluorescent emission at high concentrations of the copolymer. Confocal laser scanning microscopy (CLSM) images revealed that the FITC-labeled copolymer with higher PCL content can penetrate into a deeper layer of skin than that of copolymer with lower PCL content. The acquired data exhibit that these polymeric systems could be used as fluorescent agents in the exploration of topical drug delivery [15]. 

Maglio and co-workers synthesized a well-defined Y-shaped miktoarm star-block copolymer system of PEO and PCL. Utilizing ROP polymerization, ε-caprolactone polymerization was initiated by a lysine functionalized PEO macroinitiator and copolymers with well-defined structures were synthesized. The synthesized amphiphilic copolymers were characterized by a variety of physicochemical techniques. These copolymers self-assemble in aqueous media forming nanoaggregates where their size depends on composition. TEM imaging presented a spherical supramolecular structure with a range of sizes (100-250 nm). They developed a novel process of emulsion-melting-sonication method, which provides nanocapsules for encapsulation of a hydrophilic molecule. A fluorescent dye, rhodamine-dextran, was utilized for encapsulation and gradual release. Release rates of drug-loaded nanocapsules revealed a sharp increment at the first 4 h. Further, loaded nanocapsules exhibited a satisfying low cell toxicity, and thus this polymeric system could be a potential candidate for different drug delivery applications [16].

A novel and simple method to synthesize [PEO(PCL)_2_] star-shaped copolymers was developed by Petrova and co-workers. This group used the combination of Michael-addition reaction and ROP polymerization to synthesize first a PEO end-capped by two hydroxyl groups [PEO(OH)_2_] macroinitiator, which was utilized as the first block for the preparation of PCL blocks. The macromolecular characteristics of the copolymers were determined by size exclusion chromatography (SEC), nuclear magnetic resonance spectroscopy (^1^H-NMR), and Fourier-transform infrared spectroscopy (FT-IR). TEM observations revealed the micelle structure comprised of these copolymers as well as their sizes. Furthermore, these polymeric micelles were utilized to encapsulate the Neutral-Red (NR) and deliver it into living cells. The loaded micelles presented an accumulation in the perinuclear area of HepG2 cell lines. The obtained data revealed that the accumulation had a prolonged effect for three days, indicating the ability of these polymeric system to act as drug delivery nanocarrier [17].

Another interesting work was reported by Fan’s group, utilizing a combination of Glaser coupling and ROP polymerization (Figure 1). They synthesized an amphiphilic block copolymer consisting of [cyclic-poly(ethylene oxide) (PEO)]-b-[linear poly(3-caprolactone) (PCL)]_2_ [(c-PEO)-b-PCL_2_]. PEO block was the ring and the PCL block was the two tails of the macromolecule. ROP polymerization was used to synthesize a linear PEO with two alkyne groups at the chain terminal and two hydroxyl groups at the chain middle. Afterwards, Glaser cyclization was utilized to prepare the cyclic PEO with two hydroxyl groups. The hydroxyl groups on PEO aided the initiation of the ROP of ε-CL monomer to finally produce the (c-PEO)-b-PCL_2_ block copolymer. The synthesized copolymers were meticulously characterized by SEC, MALDI-TOF MS, ^1^H-NMR, and FT-IR [18].

Xu et al. reported a PEO-b-PPO-b-PCL amphiphilic triblock terpolymer with different ratio of hydrophobic to hydrophilic components. These triblock terpolymers were synthesized by ROP polymerization of CL initiated by the hydroxyl end group of methoxy–poly(ethylene oxide)–poly(propylene oxide) (Me–PEO–PPO) diblocks. FT-IR was able to demonstrate the polymeric structure. TEM, UV-Vis and differential scanning calorimetry revealed the self-assembly behavior in aqueous media. An interesting observation of these self-assembly nanostructures was that by increasing the PCL block length, the critical micelle concentration (CMC) was decreased significantly. Moreover, these polymeric systems were able to dissolve hydrophobic agents such as pyrene in aqueous media, while the polymeric system with a longer PCL block revealed a greater solubilizing capability. Furthermore, TEM utilization illustrated the ability to tune the size and morphology of supramolecular assemblies formed in solutions by these copolymers by varying their composition or their concentration [19].

Qi et al. developed a series of amphiphilic PEO-b-PCL diblock copolymers by ROP and anionic living polymerization. By using ROP polymerization and as initiator MePEOs with different molecular weights terminated with hydroxyl group, they could synthesize the PCL block. Moreover, they used tin octoate, (SnOct_2_) as the catalyst. Although the ROP polymerization is a common procedure to synthesize PEO-b-PCL copolymers from MePEO, the availability of MePEO hopolymers is limited. For this reason, they used anionic living polymerization of EO monomer for synthesizing PEOs of molecular weights. Afterwards, the polymerization of CL produced PEO-b-PCL copolymers with a range of molecular weights (Mn: 3.6–57 K) and PEO weight fractions (*f*PEO: 0.08–0.33). The polymerization of PEO components was made with the aid of cyanomethyl potassium as the protected initiator, which was prepared by metalation of acetonitrile with potassium naphthalenide in THF. All synthesized copolymers were characterized by ^1^H-NMR spectroscopy, DLS, and Cryo-TEM techniques [20].

Naolou and co-workers synthesized a poly(glycerol adipate)-graft-(poly(3-caprolactone)-block-poly(ethylene oxide)) (PGA-g-(PCL-b-PEO)) polymeric nanosystem. The copolymers were obtained by utilizing ROP polymerization of CL initiated by the hydroxyl group of PGA. The PCL blocks were grafted by poly(ethylene oxide) monomethylether mPEO-N_3_ utilizing copper-catalyzed azide–alkyne cycloaddition (CuAAC, “click” reaction). Moreover, for comparison, they synthesized a PEO-b-PCL diblock copolymer with the same molar mass of the grafted chains on the PGA. All synthesized copolymers self-assembled after dissolving in acetone and dialysis against water, forming polymeric micelles with sizes in the range of 10 nm. By increasing the solution temperature, the PGA-g-(PCL-b-PEO) micelles presented smaller CMC, as well as higher stability, in comparison to micelles formed by PCL-b-PEO diblocks, with molar masses similar to the grafted copolymer chain onto PGA. Their results could be useful if these systems were used as drug delivery vectors [21].

Grancharov et al. developed two polymeric nanosystems of amphiphilic (PEO-b-PCL-b-PEO) and poly(2-(dimethylamino)ethyl methacrylate)-b-poly(ε-caprolactone)-b-poly(2-(dimethylamino)ethylmethacrylate) (PDMAEMA-b-PCL-b-PDMAEMA) triblock copolymers. PEO-b-PCL-b-PEO triblock copolymer was synthesized by copper mediated “click” coupling reaction of initially modified PEO monomethyl ether and PCL-diol. Firstly, they synthesized the two outer PEO components and directly linked them to the middle PCL component by cleavable acetal groups. PDMAEMA-b-PCL-b-PDMAEMA triblock copolymer was also synthesized by atom transfer radical polymerization (ATRP). A bifunctional PCL macroinitiator was used to polymerize the DMAEMA monomer. Subsequently, the PDMAEMA were adorned with mitochondria targeting triphenylphosphonium (TPP^+^) ligands. DLS and TEM measurements were utilized to identify the micellar structure of the synthesized copolymers [22].

Another amphiphilic block copolymer system was synthesized by Liu and co-workers. Initially, they constructed a poly(ethylene oxide-co-epichlorohydrin) P(EO-co-ECH) statistical copolymer which was used to polymerize CL through the end hydroxyl group. The next step was to transform the side chlorine groups of P(EO-co-ECH) block into azido groups to synthesize the final diblock copolymer which was PCL-b-P(EO-co-glycidyl azide) (PCL-b-P(EO-co-GA)). Click chemistry was used to convert the azido group on hydrophilic block to amino or carboxylic groups. The self-assembly of these block copolymers with different pendant groups formed polymeric micelles with 96 (azido groups), 123 (amino groups), and 165 nm (acid groups) before and after conversion of side groups. Furthermore, the authors utilized a hydrophobic drug doxorubicin (DOX) to be encapsulated into the core of the polymeric micelles. Furthermore, they tested these copolymers for their cytotoxicity. The results of cytotoxicity showed good biocompatibility. Thus, these nanosystems could be useful as drug delivery nanocarriers [23].

Bhayo and co-workers designed and constructed a novel four-arm star-shaped block copolymer consisting of PEO as a hydrophilic component and PCL as a hydrophobic component. ROP polymerization was used to synthesize PCL in bulk using pentaerythritol as an initiator and stannous octoate as a catalyst (Figure 2). The next step was to use the monomethoxy poly(ethylene oxide) (MeO-PEO) and obtain the star copolymer via a coupling reaction. The synthesized copolymers were determined by ^1^H-NMR and FT-IR spectroscopy. Moreover, the SEC method was used to identify the molecular weight as well as the polydispersity of obtained copolymers [24].

Li et al. reported the synthetic procedure of PCL analogue consisting of thioether groups in the main polymeric chain by ROP polymerization of 1,4-oxathiepan-7-one (OTO). POTO was prepared by ROP polymerization at 30 °C in dichloromethane with benzyl alcohol (BnOH) as the initiator and diphenylphosphate (DPP) as the catalyst. GPC traces confirmed the well-defined POTO and POTO-b-PCL copolymers with predefined molecular weights and narrow polydispersity of molecular weights. Moreover, they used MePEO as a macroinitiator to synthesize PEO-b-POTO, PEO-b-POTO-b-PCL, and PEO-b-PCL-b-POTO, diblock copolymers, and triblock terpolymers [25].

An interesting work on the comparison of different macromolecular architectures of drug-loaded copolymers were reported by Ali and co-workers. They used a four-arm star-shaped block copolymer containing PEO and PCL components and a linear triblock copolymer (PCL-b-PEO-b-PCL). The four-arm star-shaped copolymer was the same product synthesized by Bhayo and co-workers. The linear copolymers were also synthesized by ROP polymerization of CL in bulk utilizing MeO-PEG/PEG as macro-initiator and stannous octoate [Sn(Oct)_2_] as the catalyst. The comparison of these different macromolecular architectures was made in terms of CMC, drug encapsulation efficiency, and drug release profiles. The formed polymeric micelles of both star and linear architectures were used for individual delivery of cefixime (CEF) and its co-delivery with clarithromycin (CLA). The sizes of CEF-loaded linear copolymers and of the CEF/CLA-loaded linear copolymers were 110 nm and 147 nm respectively, while sizes of the CEF-loaded star shaped copolymers and CEF/CLA-loaded star shaped copolymers were 140 nm and 173 nm respectively. Drug release profiles for the linear copolymers revealed a faster release than the star shaped copolymers. Furthermore, these copolymers were studied for their cytotoxicity utilizing three T3 cell lines. The viability trends of both architectures had similar tendency, with cell viability about 58–65%. According to the results, these synthesized copolymers could be used as effective drug delivery nanosystems [17,26].

Kulkarni et al. developed a drug delivery nanosystem for combined delivery and bioimaging application (Figure 3). Specifically, they used TPE-(OH)_2_ as a difunctional initiator and t-BuP_2_/TEB as the catalyst to synthesize copolymers via one pot consecutive organocatalytic ROP polymerization of CL and EO. Two amphiphilic block copolymers with inverse block configurations TPE-(PCL-b-PEO)_2_ and TPE-(PEO-b-PCL)_2_ were prepared by switching the consecutive addition of CL and EO. These copolymers were assembled into micelles in aqueous media by the dialysis method. The authors used two hydrophobic drugs, DOX and CUR, for their encapsulation in the hydrophobic core of the polymeric micelles. The drug release profiles revealed that in acidic conditions the drugs released more quickly than under physiological conditions. The MTT assay was utilized to evaluate the biocompatibility and cytotoxicity of these drug loaded-triblock copolymers with cancerous and normal cell lines. CUR-loaded micelles exhibited higher toxicity in cancer cell lines compared to the DOX-loaded micelles. Further, both drug-loaded polymeric micelles revealed very high biocompatibility at higher concentrations. Confocal light scattering microscopy (CLSM) allowed to verify the rapid internalization and accumulation of loaded micelles into the cells. Their important results suggest these block copolymers as potential candidates for drug delivery and intracellular bioimaging simultaneously [27].

## 3. Drug Delivery Systems

### 3.1. Micelles

The self-assembly of amphiphilic polymers in aqueous media causes the formation of micelles. Micelles are extensively used in preclinical studies for the encapsulation, delivery, and controlled release of poorly soluble APIs with several therapeutic purposes, especially for anticancer treatment. In this section, we are going to present several examples from the literature about PEO-PCL micelles and their numerous applications in drug delivery and targeting [28,29,30].

PEO-PCL micelles of different morphologies were used for the loading of paclitaxel, due to its low water solubility. The micelles were prepared by the cosolvent/evaporation method and worm-like structures were self-assembled. The spherical micelles were prepared by the sonication of the worm-like fillomicelles, while their size remained constant. Their size was around 50 nm, and after one month their size was increased to 150 nm, as temporal stability studies revealed. The loading of the API did not change the size distribution of the spherical micelles. For the evaluation of the API-loaded micelles cytotoxicity, studies were performed on A549 human lung carcinoma cells. The authors found that the spherical structures of PEO-PCL micelles showed an ameliorated nanotoxicity profile and higher efficacy in targeting the cytotoxic dose of paclitaxel in tumour side [31].

Poly(ethylene oxide)-block-poly(propylene oxide)-block-poly(ϵ-caprolactone) (PEO-PPO-PCL) were also synthesized for creating micelles to co-deliver the antitumor API docetaxel and the autophagy inhibitor chloroquine. The D-α-tocopheryl poly(ethylene glycol) (TPGS) was also used for the self-assembly of PEO-PPO-PCL into micellar structures. The micelles showed sizes at the nanoscale and sustained release kinetics of the encapsulated APIs. The prepared nanomicelles can easily penetrate cancer cells with ameliorated efficacy, biocompatibility, and therapeutic profile, as the in vitro cytotoxicity experiments showed [32].

Rituximab-poly(ethylene glycol)-1,2-distearoyl-*sn*-glycero-3-phosphoethanolamine was encapsulated into micelles of methoxy-PEO-PCL or methoxy poly(ethylene oxide)-poly(ε-benzylcarboxylate-ε-caprolactone) (PEO-PBCL) micelles for targeting B-cell lymphoma cells. These systems prepared by the nanoprecipitation method were immunomicelles with size of both species lower than 100 nm, with great kinetic and thermodynamic stability. The PEO-PCL micelles showed a sustained release profile for 50 h [33].

From the same group, PEO-PCL micelles were developed for targeting to breast cancer calls. The authors also studied the effect of the core structure on the tumour accumulation. Namely, micellar stabilization using PCL cores enhanced the level and duration of micellar structure accumulation in breast tumor tissues [34].

Micelles based on amphiphilic PCL-PEO triblock and star-shaped diblock copolymers were prepared by the dialysis method. Their size was lower than 50 nm and exhibited slightly negative zeta potential. The biocompatibility of the prepared micellar structures was further evaluated by using red blood cells and HeLa cells. The results showed that these nanomicelles, with components of linear or star architectures, showed great hemocompatibility at concentrations lower than CMC. According to the authors, these micelles are ideal carriers for the delivery of hydrophobic APIs, especially those with the star-shaped architecture [35].

PEO-PCL micelles and polymersomes were studied for their internalization into tumors and endothelial cells in conventional 2D monolayers and 3D tumor spheroids. The internalization of the encapsulated dye was faster and higher with the micellar structures in comparison to the polymersomes prepared by the same block copolymer, probably due to their smaller size. The surface characteristics of both nanocarriers were similar [13].

Molecular PCL-PEO worms were prepared from linear and core/shell wormlike polymer brushes loaded with doxorubicin, via hydrophobic interactions towards PCL chains. The last micellar system showed lower encapsulation efficiency and a faster release profile of the doxorubicin drug, as well as higher penetration into cells (Figure 4) [36].

PEO-b-PCL was also used as excipient for the encapsulation of the antibiotic cyclosporine A. Different micelles were prepared by the co-solvent evaporation method with the same hydrophilic PEO chains combined by PCL chains with three different molecular weights. The size of the micelles increased with the increase of the molecular weight of the PCL chain but, in all cases, the size of the prepared micelles remained below the 100 nm. The marketed product of cyclosporine A was used for comparison reasons. The solubilization of the encapsulated API and its release profile were found to be strongly dependent on the composition of the block copolymer. The controlled release profile of the cyclosporine was achieved by the micelles after intravenous administration, probably due to the structural properties of PEO-b-PCL in comparison to the low molecular weight of the surfactant of the commercial pharmaceutical product [37].

Two studies for the further evaluation of cyclosporine A loaded PEO-b-PCL have been already published in the literature the last five years. The first one studied the pharmacokinetics of the orally administered micellar carriers in rats versus the marketed pharmaceutical product Neoral^®^ (microemulsion). The micelles showed high stability to the biorelevant media used to mimic the conditions of the gastrointestinal track. The results showed that the C_max_ of micellar cyclosporin A was significantly higher than that obtained from the microemulsion formulation, and the API exhibited higher blood-to-plasma amounts, too. The second one investigated both the pharmacokinetics and tissue distribution in rates in comparison to the commercial formulation Sandimmune^®^ (which is a suspension composed of surfactant). The two formulations showed more or less the same pharmacokinetic profile. On the other hand, the tissue distribution was quite different. Namely, the PEO-b-PCL formulation showed higher AUC of cyclosporin A in liver, lower in spleen, lungs, and kidneys and comparable in heart. The last means that the selection of the excipient for the development of a nanoformulation strongly affects the distribution of the encapsulated API by altering the biological interactions with the specific tissues [38,39].

PEO-PCL micelles were also used for improving the ADMET profile of curcumin. The micellar carriers without and with the selected API were fabricated by the cosolvent evaporation technique. The molecular weight of the block copolymer and the ratio of the curcumin/polymer played a key role in determining the size and size distribution of the micelles, as well as the encapsulation efficiency. The micellar systems exhibited sizes between 50 to 200 nm and encapsulation efficiency ranged from 5% to 60%. The most efficient carrier for the solubilization of API was prepared by PEO(5000)-PCL(24500). The best carrier for the controlled release profile of curcumin was prepared by PEO(5000)-PCL(13000). Furthermore, the micellar curcumin was found to be effective in different tumour cell lines, without any stability limitation due to hydrolytic degradation in physiological conditions [40].

The in vitro cytotoxicity profile of chemically conjugated doxorubicin into PEO-b-PCL nanocarriers was evaluated in different cellular cultures, having added value due to the hydrolyzable PCL cores of the micellar carriers. The last property was also found to be important for the release of doxorubicin in biorelevant media mimicking the conditions of tumour tissues [41].

Xiong et al. synthesized and characterized PEO-b-PCL based copolymers with polyamine side chains on the PCL polymeric chain, i.e., with grafted spermine or tetraethylenepentamine or N,N-dimethyldipropylenetriamine. Their biocompatibility profile was more or less the same as the pure counterpart, without any further functionalization, PEO-b-PCL copolymer. These micellar carriers were loaded with siRNA and showed effective endosomal escape after their uptake from cells [42].

The uptake of PEO-PCL micelles by human breast cancer cells is strongly dependent on the length of both polymeric chains. Namely, the PEO chain should exhibit molecular weight around 5000 g/mol, while the PCL around 13,000 g/mol. The last is very important because one can control the biological interactions of PEO-PCL by controlling the molecular characteristics and composition of the block copolymer [43].

PEO-PCL micelles have been already used for the solubilization and amelioration of the pharmacokinetic profile of valspodar, which is a hydrophobic compound and a non-competitive inhibitor of P-glycoprotein. This API has been developed to overcome multi-drug resistance, a phenomenon common to a wide range of chemotherapeutics. The prepared micelles presented sizes below 70 nm and the encapsulation efficiency was found to be more than 90%. The plasma pharmacokinetic parameters of valspodar in rats following a single i.v. administration (5 mg/kg) and a single oral administration (10 mg/kg) were found to be improved in comparison to the same API in Cremophor EL^®^. The most important results showed that these nanomicellar carriers can improve the pharmacokinetic profile of the API in rats after i.v. and oral administration by decreasing the clearance rate of the encapsulated API [44].

Recently, the PEO-PCL block copolymer was modified with different ligands, i.e., spermine (a polyamine), TAT ligand and N-[N-[(S)-1,3-dicarboxypropyl] carbamoyl]-(S)-lysine (DCL) to target prostate cancer. The prepared multifunctional nanoparticles can carry the anticancer API docetaxel and anti-nucleostemin siRNA. It should be pointed out that the biocompatibility profile of the pure micellar carriers was very good, as the cell viability experiments revealed. A prolonged release profile of the encapsulated anticancer API was observed after the degradation of the micellar carrier [45].

PCL-b-PEO was also conjugated with the cyclic peptide Arginine-Glycine-Aspartic acid-d-Phenylalanine-Lysine (c(RGDfK)) and fluorescein isothiocyanate (FITC). The prepared nanomicelles were loaded with the anticancer agent doxorubicin for intravesical instilled chemotherapy of superficial bladder cancer. The results showed that there was strong affinity and inhibitor affinity of the conjugated nanomicelles to bladder cancer cells [46].

In the same context, PEO-b-PCL micelles were decorated on their surface with alphavbeta3 integrin-targeting ligand (i.e., RGD4C) in order to ameliorate the therapeutic index of doxorubicin for targeting sensitive and resistant tumour tissues. Another formulation was also developed by conjugating the doxorubicin to the core of the micelles using the more stable amide bonds. The surface modification increased the cellular uptake of the encapsulated API to sensitive cancer cell lines, accompanied by the accumulation of doxorubicin in the mitochondria (targeting to sub-cellular organelles). The above is one more example from the literature proving that PEO-PCL nanomicelles could be a multifunctional system for the treatment of different types of cancers [47].

Furthermore, PEO-PCL micelles prepared by co-solvent extraction method were loaded with paclitaxel prodrugs. The results showed that the solubility of these compound was ameliorated by more than five times and the size of the nanocarriers was found below 50 nm. Sustained release from the nanomicellar carriers was observed for all the studied compounds and all the pharmacokinetic parameters were found to be ameliorated. Namely, an increase in area under the curve, half-life, and mean residence time was observed while total clearance and volume of distribution was decreased [48].

The behaviour of PEO-PCL micelles derived from copolymers with linear triblock and four-arm star-diblock architectures for the delivery of docetaxel was also investigated. Both prepared micelle formulations exhibited more or less the same physicochemical characteristics and loading and release profiles of the encapsulated docetaxel (sustained release for more that fifteen days). On the other hand, the stability of micelles composed of four-arm star-diblock architectures was limited in conditions mimicking the human plasma. Micelles composed of linear triblock polymeric chains were less cytotoxic in G(2)/M phase synchronized cells. The authors concluded that the architecture of the polymeric chains is one of the most important parameters that should be taken into consideration for the design and the development of PEO-PCL nanomicellar formulations [40,49].

Phosphonium-functionalized PEO-PCL micelles were developed and studied for their antibacterial activity. According to the findings, the minimum bactericidal concentration depended on the phosphonium alkyl chain length, and different trends were observed for Gram-(-) and Gram-(+) bacteria. In the same system, the antibiotic agent tetracycline was successfully incorporated, leading to a potential multimechanistic nanosystem for infectious diseases [50].

A physicochemical study, in which isobaric-isothermal molecular dynamics simulation was included, was conducted by loading linear and branched PEO-PCL block copolymers with the hydrophobic API cucurbitacin B. The chain mobility and the degree of swelling were found to be dependent on the interactions between the API and the PCL chains. These studies could be a road map for the amelioration of the loading efficiency and the release profile of hydrophobic API from PEO-PCL nanomicelles [51,52].

Another important application of PEO-PCL block copolymers is for imaging purposes. For this reason, Park et al. conjugated diamino-PEO both with a PCL chain and a ligand for specific radioisotope. The CMC of the conjugated PEO-PCL was at 25 mg/mL and their size at 60 nm, which were ideal characteristics for blood vessel and bone imaging. The stability in human serum and the extremely high effectiveness of the labelled polymeric nanomicelles, accompanied by limited liver and spleen uptake, led to the outcome that this system could be a potential candidate for diagnostic and imaging applications [53].

### 3.2. Polymeric Nanoparticles

An interesting approach for development of a drug delivery nanosystem containing PEO-b-PCL block copolymer was reported by Chen et al. Specifically, they utilized a two-phase gas-liquid microfluidic reactor to manipulate the structure and characteristics of drug-loaded block copolymers. Curcumin (CUR) was utilized as a hydrophobic agent to be encapsulated in different feed ratios into block copolymer nanostructures. CUR-loaded copolymers were prepared using two different procedures. The bulk method presented a reduced encapsulation efficiency and increased drug precipitation as the loading ratio rises. In contrast, the drug-loaded block copolymer constructed by microfluidic manufacturing reveal encapsulation efficiency and drug loading that both rise as the drug to copolymer ratio increases. The drug release profiles exhibited a fast release of the drug in the first 5 h followed by sustained release. Furthermore, they used the MDA-MB-231 cancer lines to investigate the cytotoxicity of the prepared CUR-loaded nanoparticles. They found no significant cytotoxicity for the loaded nanoparticles, which means that these nanosystems could be used as drug delivery systems with controllable structure and properties [54].

Mirzaghavami et al. [55] conjugated folic acid to a magnetite (SPION) PEG- PCL-PEG triblock copolymer, forming polymeric nanoparticles, being loaded with 5-Fluorouracil for targeted delivery of drug to HT29 colon cancer cells. The preparation of the 5-FU loaded magnetite/PEG-PCL-PEG-Folic nanoparticles was performed by using the double emulsion solvent evaporation method. The use of the folate-targeting nanoparticles resulted to improvement in therapeutic index of 5-Fluorouracil, better antitumor efficiency percentage of cell death, ROS production, and colony formation ability, according to the obtained in vitro data from various cell types.

PEG-PCL biodegradable polymersomes were developed by Pang et al. [56], being conjugated with mouse-anti-rat monoclonal antibody OX26 (OX26-PO), as a novel brain drug delivery system. According to the morphological and physiochemical data obtained by electron microscopy and light scattering techniques, the OX26-PO had a round and vesicle-like shape with a mean diameter around 100 nm. Coupling of OX26 with PO was confirmed by immuno-gold labeling, while the surface OX26 densities were obtained from enzyme-linked immunosorbant assay. Furthermore, a model peptide, NC-1900, was encapsulated into OX26_34_-PO. OX26_34_-PO significantly enhanced the brain delivery of NC-1900 with ameliorating the scopolamine-induced learning and memory impairments via i.v. administration, proving a new path to the peptide delivery directly to the central nervous system. Taking into account these findings, the simplicity, sensitivity, and speed of the nanoenzyme technique are its benefits, whereas its cost, point-of-care needs for accuracy and reaction time are its drawbacks.

In another study, a biodegradable triblock copolymer PCL-PEG-PCL (PCEC) was synthesized by the ring-opening polymerization method and polymeric nanoparticles were formed by the one-step modified emulsion solvent evaporation method. The obtained cationic PCEC nanoparticle was employed to condense and adsorb DNA onto its surface and more specifically the model plasmid GFP (pGFP). The DNA-nanoparticles weight ratio strongly affected the size of the nanoparticles in an analogous trend. Thus, there is a ratio limitation in order to avoid large aggregates. According to the obtained results, the PCL-PEG-PCL exhibits a great potential in DNA delivery [57].

Liu et al. [58] designed “intelligent” polymeric nanoparticles, consisting of a gelatinase-cleavage peptide with PEG and PCL-based structure, for tumor-targeted docetaxel delivery (DOC-TNPs). The gelatinase-stimuli PEG-Pep-PCL nanoparticles were prepared by inserting the enzyme-specific substrate between PEG and PCL blocks. Compared with pure PEG-PCL nanoparticles, the gelatinase-ones overcame some limitations of PEGylation, provoked specific and sufficient interactions with cancer cells, maximum nanoparticles distribution, insertion to the tumor cells, and avoidance of fast elimination from the tumor tissue. Most importantly, the DOC-TNPs showed higher antitumor efficacy and lower toxicity than Taxotere^®^. The authors concluded that these data indicate a promising strategy to improve clinical cancer therapy while minimizing side effects in various overexpressed gelatinase cancers.

Grossen et al. [59] designed solid-sphere nanoparticles (SNPs) from PEG-b-PCL and covalently conjugated the monoclonal antibody (83-14 mAb) to their surface by a PEG-spaced heterobifunctional linker. mAb targets against the human insulin receptor and is highly expressed on human brain microvascular endothelial cells. Apart from the physicochemical and morphological characteristics of the nanoparticles, their interactions with eukaryotic cells and potential toxic effects were studied in vitro in a BBB model (hCMEC/D3) and the human hepatocellular carcinoma cell line HepG2. According to the results, there was a successful antibody conjugation, along with sufficient cellular uptake crossing the blood–brain barrier and without significant toxicity, being thus suitable for drug delivery to the central nervous system.

In another case of polymersomes, Zou et al. [60] functionalized PEG-b-PCL polymersomes with nanobodies, prepared by thin film hydration and nanoprecipitation methods and confirmed by cryo-TEM. The nanoparticles were functionalized with either anti-HER2 or anti-GFP nanobodies using maleimide-cysteine chemistry and were treated to HER2 positive breast cancer cells. Cryo-TEM revealed that simple vesicles co-existed with vesicles-in-vesicles structures and spherical micelles, while some rod-shape nano-objects were observed after the nanobody functionalization. According to the results, the anti-HER2-functionalized PEO-b-PCL polymersomes are able to specifically target breast cancer cells expressing HER2 receptors.

Gu et al. [61] utilized an activatable low molecular weight protamine (ALMWP) and conjugated it to PEG-PCL nanoparticles. The fabricated nanoparticles were designed for enhanced targeted glioblastoma therapy. The ALMWP-NP were loaded with paclitaxel (PTX). Desirable pharmacokinetic and biodistribution profiles for anti-glioblastoma drug delivery were observed by using in vitro assays and in vivo imaging. More specifically an elevated metalloproteinases-dependent cellular accumulation in C6 cells was observed, due to enhanced glioma targeting ability and tumor penetrating ability, as well as improved cytotoxicity of PTX, compared to the market approved paclitaxel (Taxol^®^).

In another case study of brain glioma, Xin et al. [62] developed a dual-targeting nanoparticle by conjugating Angiopep with PEG-PCL nanoparticles (ANG-NP), as an effort to enhance the nanoparticulate penetration across the blood–brain barrier (BBB) and into the tumor tissue during anticancer therapies. Angiopep-2 is a ligand of LRP that possesses a high brain penetration capability of the BBB. ANG-NP were designed to target the low-density lipoprotein receptor-related protein (LRP), being over-expressed on the BBB and glioma cells. The fabricated nanoparticles were labeled with rhodamine and loaded with paclitaxel. Angiopep-conjugated PEG-PCL nanoparticles exhibited increased tumor penetration, accumulation, and successful decrease of tumor cells.

In another anticancer formulation containing doxorubicin, PEG-PCL core-corona nanoparticles were employed and decorated with hyaluronic acid (HA-PEG-PCL). The nanoparticles were characterized in morphological and physicochemical terms, drug entrapment, and in vitro drug release profile. The HA-PEG-PCL nanoparticles could release DOX for up to 17 days in a sustained release manner. The nanoparticles were disposed by intravenous injection in Ehrlich ascites tumor (EAT)-bearing mice, revealing successful tissue distribution and tumor growth inhibition [63].

Mannan functionalized anionic PCL-PEG-PCL nanoparticles were prepared by emulsion solvent evaporation method at one step. Human basic fibroblast growth factor (bFGF), being a cationic protein, was absorbed onto anionic polymeric nanoparticles surface due to electrostatic interaction. The results indicated that there were increased autoantibody IgG, IgG1, and IgG2a titers in the mice immunized by the mannan modified PCL-PEG-PCL polymeric nanoparticles, due to the mannan targeting ability to dendritic cells (DCs), thus improving humoral immunity. Authors concluded that the prepared nanoparticles exhibit a great application potential as vaccine delivery systems [64].

Lactoferrin (Lf) has also been reported as decoration to PEG-PCL nanoparticles by Liu et al. [65]. Lactoferrin is utilized to facilitate the nose-to-brain drug delivery of neuroprotection peptides, enabling non-invasive drug delivery in treatment of Alzheimer’s disease. Indeed, the Lf-conjugated PEG-PCL nanoparticle (Lf-NP) performed significantly enhanced cellular accumulation in vitro. After intranasal administration in rats, there was higher distribution of nanoparticles in the brain. The treatment with a neuroprotective peptide, carried by the nanoparticles, resulted in a significant improvement of neuroprotection and memory in both behavioral and histological studies.

Surnar et al. [66] designed a dual drug delivery pH responsive vehicle for oral administration via the gastrointestinal tract, employing functionalized PCL block copolymer and more specifically a group of amphiphilic diblocks, PEG-b-CPCL_x_, with x = 25, 50, 75, and 100 that were able to self-assemble in vesicles of 100–250 nm diameter. The polymeric vesicles were capable of multiple loading, with both hydrophilic molecules in the core and hydrophobic drugs in the layer. Moreover, the vesicles exhibited pH-response, being stable in strong acidic conditions (pH < 2.0, stomach) but ruptured under neutral or basic pH (7.0 ≤ pH, similar to that of small intestine), and thus able to trigger content release in the intestine where the absorption process is taking place, leading to increased bioavailability. Different mechanisms of release kinetics were followed, depending on the formulation of the vesicle.

### 3.3. Hybrid Polymer-Lipid Nanopartilces

Polymer-lipid hybrid nanoparticles combine the advantages of both polymeric and lipidic nanoparticles, including liposomes, niosomes, and lyotropic liquid crystalline nanosystems. These hybrid systems overcome the limitations of lipids and polymers and offer enormous potential in the field of nanomedicine. As defined by Mohanty et al. [67] polymer-lipid hybrid nanoparticles contain three major components: (1) a hydrophobic/hydrophilic polymeric core encapsulating both hydrophobic and hydrophilic drugs effectively and resulting in sustained release kinetics; (2) a lipid shell surrounding the polymeric core with high biocompatibility, overall stability, and reduced retention of the drug inside the polymeric core; (3) and an outer component, consisting of a lipid-polyethylene glycol (PEG), which is covered by a lipid layer to enhance steric stabilization, prolong circulation time, and prevent immune recognition.

Pippa and co-workers designed and developed a novel chimeric (hybrid) amphiphilic system containing DPPC (dipalmitoylphosphatidylcholine) and PEO-b-PCL diblock copolymer. The self-assembly in aqueous and biological medium of synthesized nanosystems was studied by dynamic, static, and electrophoretic light scattering to determine the structure, morphology, size and the surface charge of them. All of these characteristics depend on the block copolymer content, temperature, and concentration. The embedding of the block copolymer resulted in smaller sizes of the nano-aggregates. The chimeric nanosystems retained their initial structural characteristics for two weeks. Their sizes decreased by increasing temperature until 50 °C. The microenvironment of these nanoparticles presented changes in HPLC-grade water and PBS by increasing the polymer component. Moreover, these nanosystems were utilized to incorporate the hydrophobic drug indomethacin (IND). The IND-loaded nanoparticles revealed a reduction of the size. Furthermore, the drug-loaded mixed systems exhibit a greater incorporation efficiency in PBS than in HPLC-grade water. Their obtained data showed that these chimeric nanosystems could be potential candidates for the entrapment and delivery of IND [68].

More recently Pippa et al. designed a drug delivery nanosystem composed mainly of non-ionic surfactants and cholesterol. Specifically, they constructed mixed systems of PEO-b-PCL diblock copolymer, tween 80 or span 80 or cholesterol with different molar ratios of the components. They studied the self-assembly character of neat surfactant/cholesterol and mixed surfactant:cholesterol:PEO-b-PCL systems. The self-assembly protocol was the thin-film hydration method. The physicochemical behavior was studied by a plethora of physicochemical methods such as light scattering techniques and cryogenic transmission electron microscopy (Cryo-TEM). The acquired data showed that the morphology and the sizes of neat surfactant/cholesterol niosomes were affected by the presence of block copolymer. In vitro experiments revealed a significantly low cytotoxicity of the prepared nanosystems. All the systems presented good colloidal stability, which makes these systems suitable nanocarriers for a variety of biomedical applications [69].

Zhang et al. [70] developed a novel lipid-polymer hybrid drug carrier comprised of folate modified lipid monolayer shell and polymer-core nanoparticles for the sustained, controlled, and targeted delivery of paclitaxel (PTX) for tumor-targeted therapy. The core-shell nanoparticles consisted of a PCL hydrophobic core based on the self-assembly of PCL-PEG-PCL amphiphilic copolymers, a lipid monolayer formed with 1,2-distearoyl-sn-glycero-3-phosphoethanolamine-N-[methoxy (polyethylene glycol)-2000] (DSPE-PEG2000), and a targeting ligand (folate) on the surface. According to the results, the prepared hybrid nanosystems exhibited successful internalization efficiency and targeting ability to the EMT6 breast cancer cells, which overexpress folate receptor. Most significantly, the cytotoxic effect of PTX-loaded hybrid nanosystems was lower than that of Taxol(^®^), indicating lower toxicity, but with similar antitumor efficacy.

Papagiannopoulos et al. [71] investigated the ability of the combination of the DPPC lipid and of the amphiphilic diblock copolymers PEO-b-PCL to stabilize uni-lamellar nano-vesicles. Two different PEO-b-PCL copolymers were used, exhibiting different PCL content. Small angle neutron scattering (SANS) was used to define their size distribution and bilayer structure and resolve the copresence of aggregates and clusters in solution. The hybrid nanosystems exhibited a broad size distribution, indicating the presence of bilayer membranes of relatively low bending stiffness. Moreover, there was a temperature-analogous increase of their mean diameter, while their number density and mass were higher in the case of the diblock copolymer with the larger hydrophobic block, highlighting the development of lipid-block copolymer vesicles with controlled lamellarity, being suitable for drug delivery.

The fluidity of the pure liposome formulations hinders adhesion to the wound surface. Madecassoside exhibits excellent therapeutic effects in wound healing and scar management. However, due to its high hydrophilic nature, there is low permeability through skin tissue that limits its topical application. Liu et al. [72] formulated liposomes with madecassonide by employing a biodegradable and temperature-responsive PEG-PCL-PEG copolymer, as a regulator of the adhesion properties of the liposomes. The hybrid liposomes were topically administrated and were maintained in a hydrogel state for better adhesion until the temperature reached 43 °C, enabling better adhesion. They exhibited superior wound contraction and healing effects relative to the conventional madecassoside liposomes in second-degree burn experiments.

Gibot et al. [73] investigated the micelle–membrane interactions of PEO-PCL copolymer and poly(ethylene oxide)-block-poly styrene, PEO-PS, on biomimetic membranes and also carried out biological experiments on two-dimensional (2D) and three-dimensional (3D) cell cultures, evaluating the uptake of a photosensitizer, Pheophorbide a (Pheo). Pheo loaded PEO-PCL micelles allowed a greater cellular uptake of photosensitizer than Pheo loaded PEO-PS micelles, but with lower cell viability. The authors concluded that PEO-PCL micelles could be incorporated into membranes, leading to optimized photosensitizer activity and improvement of photodynamic therapy efficiency.

He et al. [74] fabricated novel stealth nanoparticles, composed of PEG-b-PCL, soybean phosphatidylcholine (SPC), and cholesterol, for paclitaxel (PTX) delivery, in an effort to improve the cellular uptake of already existing PEGylated liposomes. Two PEG-b-PCL polymers with different molecular weights were used to fabricate stealth nanoparticles and were compared with conventional PEGylated liposomes of SPC, cholesterol, and DSPE-PEG2000. The physical properties, cellular uptake, endocytosis pathway, cytotoxicity, pharmacokinetics, tumor accumulation, and anticancer efficacy were evaluated in vivo after injection into 4T1 breast tumor-bearing mice. The PCL modification of the PEGylated liposomes increased drug accumulation at the target tumor site and tumor cells, exhibiting promising results towards new stealth-based nanoparticles with anticancer activity.

Khan et al. [75] used low-molecular-weight (of 1.25–3.45 kDa) biodegradable block copolymers, including PEG-PCL, to construct nano- and micron-scaled hybrid polymer-lipid vesicles. The physicochemical characteristics were evaluated, regarding size, bilayer thickness, small molecule encapsulation, surface topography, and self-assembly ability. It was found that the low molecular weight polymers cannot form well defined polymer vesicles as polymersomes in the absence of the phospholipid. In the presence of the phospholipids, there was a homogeneous bilayer thickness similar to lipid vesicles, but with a variability in the surface topology (Figure 5). Moreover, there was a phospholipase-dependent sensitivity, being affected by the type of the polymer.

DPPC lipid has also been combined with triblock copolymers of the PCL_n_-PEO_m_-PCL_n_ type, where n = 12 and m = 45 or n = 16 and m = 104, to form mixed hybrid polymer lipid vesicles. According to the physicochemical characterization, the copolymers affected the interface of the lipid bilayer and increased the main phase transition, pointing to an increase in the thermodynamic stability as well as in the order of the phospholipid acyl chain packing in the gel phase. The authors concluded that the copolymers were predominantly inserted into the bilayer, but without causing the collapse of the bilayer. The amount of the hydrophilic PEO differentiated the results between the two copolymers [76].

Chountoulesi et al. [77] utilized two different PEO-b-PCL block copolymers of different PCL content as stabilizers in lyotropic liquid crystalline nanoparticles prepared from glyceryl monooleate lipid. In this study, the typically used amphiphilic stabilizer in liquid crystals Pluronic^®^ F-127/Poloxamer P407 was replaced by PEO-PCL. Physicochemical, morphological, and thermal evaluation was carried out by light scattering, cryo-TEM, and micro-differential scanning calorimetry methods. The amount of PCL in the block copolymer differentiated the characteristics of the nanosystems. For example, PEO-b-PCL with 15 wt% PCL resulted in highly organized cubic nanoparticles of primitive type bicontinuous cubic phase and multilamellar striated nanoparticles, while PEO-b-PCL with 30 wt% PCL resulted in mixed lipid-polymer micelles with worm-like morphology. Subsequently, resveratrol was loaded onto the nanosystems, and in vitro release studies were carried out, revealing a strong dependence of the drug release kinetics with the morphology of the nanoparticles, and thus with the type of the polymeric stabilizer.

Most recently, Chountoulesi et al. [78] developed another group of lyotropic liquid crystalline nanoparticles of glyceryl monooleate lipid that were mainly stabilized by the poly(2-(light scattering, cryo-TEM and micro-Differential Scanning Calorimetry methods, while dimethylamino)ethyl methacrylate)-*b*-poly(lauryl methacrylate) block copolymer carrying tri-phenyl-phosphine cations (TPP-QPDMAEMA-*b*-PLMA). In these formulations, other co-stabilizers were also tried, including PEO-b-PCL copolymers. The systems were found to perform simultaneously sub-cellular targeting, stimuli-responsiveness, and stealthiness showing a stimuli-triggered (pH and temperature) and controlled drug release profile. Physicochemical, morphological, and thermal evaluation was carried out by the subcellular localization was monitored by confocal microscopy, revealing targeting of lysosomes. The combination of TPP-QPDMAEMA-*b*-PLMA and PEO-b-PCL copolymers resulted in the formation of liquid crystalline nanoparticles exhibiting a special morphology with a highly ordered inner structure with a core covered by intersecting lamellas.

## 4. Technology of PEO-PCL Based Nanosystems for Drug Delivery

As mentioned above, the supramolecular structures of copolymers strongly depend on the ratio of blocks in macromolecules and, for this reason, there is an interdependence between the macromolecule-to-supramolecular structure. Namely, PEO-PCL linear and core/shell worm-like brushes showed different %encapsulation efficiencies of low-molecular weight APIs, like doxorubicin. The worm-like brushes exhibited 26% drug loading while the spherical micelles around 50% [36]. Additionally, the solubilization of hydrophobic APIs, like curcumin is also dependent on the ratio of the blocks of PEO-PCL. PEO(5000)-PCL(24500) was found to be the best copolymer, achieving the highest solubilization of curcumin [40]. For polymersomes, the DNA-PCL-PEG-PCL weight ratio strongly affected the physicochemical characteristics of the supramolecular structures and there was a ratio limitation in order to avoid large aggregates [57]. The PCL ratio in the group of amphiphilic diblocks, PEG-b-CPCL_x_, with x = 25, 50, 75, and 100 controlled the encapsulation efficiency of APIs and the pH-responsive release in the different compartments of the gastrointestinal tract [66]. The stability and the release profile of hybrid PEO-PCL nanoparticles is also strongly affected by the molar ratio between the components, i.e., lipid:PEO-PCL ratio. Generally, a higher ratio of the polymeric guest stabilized the characteristics of the lipid bilayer and delayed the release of hydrophobic APIs [68,71].

## 5. Conclusions

PEO-PCL block or graft copolymers, belonging to the family of biocompatible and biodegradable copolymer and exhibiting stealth properties, due to the presence of PEO blocks, show great potential in biomedical applications and most interestingly as drug delivery nanocarriers (Figure 1 and Figure 2). As depicted by the studies described above, the employment of PEO-PCL in nanocarriers, such as micelles and polymeric or hybrid lipid-polymeric nanoparticles, provides safer and more efficient drug treatments, by improving drug biodistribution and tissue penetration, also increasing the bioavailability and decreasing the toxicity and side effects. As already indicated, PEO-PCL and consequently PEO-PCL-based nanoparticles offer flexibility in terms of design from a technological standpoint. Initially, these polymers come in a variety of kinds (compositions and architectures). These nanosystem physicochemical properties are essential to how they behave both in vitro and in vivo. All the techniques for their characterization are summarized in Table 1. It is also possible to create controlled release qualities and targeting by using various functionalization techniques. Moreover, PEO-PCL can be easily functionalized, allowing targeting ability. On the other hand, there are just a few products/medicines where PEO-PCL copolymers are utilized. All the nanomedicines and PEO-PCL formulations in development must undergo a thorough examination in terms of the physicochemical and morphological characteristics utilizing specialist procedures. Preclinical research is more expensive because of this need compared to other pharmaceutical formulations, i.e., tablets, suspensions, etc. At every stage of the design and development of the aforementioned platforms, as well as for knowledge transfer between academic institutions and the pharmaceutical companies, big pharma also needs scientists (chemists and formulation scientists) with training and great experience in the field of the synthesis and characterization of PEO-PCL materials and formulations.

In conclusion, we have discussed a number of PEO-PCL drug delivery nanosystem examples from the literature that have emerged in recent years at various preclinical and clinical study stages. For the continued development of these systems as nanomedicines, the results and outcomes were in the majority of the stated situations highly positive. However, the synthesis and the particular molecular characteristics of the copolymer can affect the characteristics of the respective nanocarrier, e.g., size, morphology, and drug loading efficiency. Thus, a careful design of polymer synthesis and nanocarrier fabrication should be carried out to achieve better performance of the nanocarrier systems.

## Data Availability

Not applicable.

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
