# Peer review of "Recent Advances on PEO-PCL Block and Graft Copolymers as Nanocarriers for Drug Delivery Applications"

_materials, 2023, doi:10.3390/ma16062298_

Round 1

Reviewer 1 Report

The review is dedicated to copolymers of ethyleneoxide and caprolactone as drug delivery vehicles. The information about application of copolymers as nanocarriers is sufficient but some information should be added prior acception of the manuscript.

1)Please add more references into introduction section.

2)Supramolecular structures of copolymers strongly depend on ratio of blocks in macromolecules. I suggest to add a paragraph dedicated to quantitative analysis of composition of macromolecule - to - supramolecular structure relation.

Author Response

Reviewer #1

Comment

1)Please add more references into introduction section.

Answer

According to the reviewer’s comment, we enriched the introduction with a new paragraph and 9 more references, as follows:

Polymer materials are widely used in the field of pharmaceutical technology for the design and development of different carriers for the delivery and targeting of Active Pharmaceutical Ingredients (APIs). In the recent years, polymer materials are of paramount importance for the nanodelivery as drug delivery platforms of anticancer APIs, proteins, antioxidants, antigens for vaccine delivery as well as nanoreactors and artificial organelles [1-5].

Poly(ethylene oxide)-poly(É›-caprolactone) (PEO-PCL) is a family of block (or graft) copolymers with several biomedical applications. These types of copolymers are well-known for good biocompatibility and biodegradability properties, being ideal for biomedical applications and being used for the formation of a variety of nanosystems intended for controlled drug release [6-11]. They exhibit also great potential in tissue engineering and medicinal chemistry, too.  Several types of particles can be formed by using PEO-PCL block or graft copolymers, ranging from nano- to micro-structures. In this review, we will focus only on the nanocarriers formed by the utilization of PEO-PCL copolymers. In Scheme 1, the structure and the chemical architecture of PEO-PCL block or graft polymers as well as the different structures than can be obtained by the self-assembly and the formulation techniques are illustrated. PEO is a hydrophilic, non-ionic block with low immunogenicity and high blood compatibility. It is already used for modified the surface of liposomal marketed nanomedicines. Due to its hydrophilicity, the interactions with plasma proteins are limited, forming “stealth” nanoparticles. On the other hand, PCL is a hydrophobic, biocompatible and biodegradable polymer with semi-crystalline properties [6-11].   PCL based formulations have been already used for tissue engineering applications, too [12].

The PEO-b-PCL block copolymers are used in drug delivery because they can ameliorate the Absorption, Distribution, Metabolism, Excretion and Toxicology (ADMET) profile of the encapsulated Active Pharmaceutical Ingredients (APIs). They can be used for active targeting to improve drug cellular internalization or for passive targeting (Enhanced Permeability and Retention- EPR effect). Last but not least, the degradability of PCL block in acidic pH, gives the opportunity to formulation scientists to explore triggered drug release mechanisms. The stealth properties of the nanostructures self-assembled by PEO-PCL copolymers have been reported by in vitro and in vivo studies. The formation of a protein corona is limited due to the hydrophilic PEO block. The last property is very important for extended circulation times in the human body by intravenous (iv) administration. Except from drug delivery purposes, PEO-PCL have been already used for imaging and diagnostic purposes.  Low and high molecular weight APIs have been encapsulated in PEO-PCL nanocarriers by different methods, i.e., incorporation into the PCL core, conjugation in the PEO polymeric chains, etc. [13].

In the reference section, we added:

  1. Pippa, N.; Pispas, S.; Demetzos, D. Polymer Self-assembled nanostructures as innovative drug nanocarrier platforms. Curr Pharm Des 2016, 22, 2788-95.
  2. Tanner, P.; Baumann, P.; Enea, R.; Onaca, O.; Palivan, C.; Meier, Polymeric vesicles: from drug carriers to nanoreactors and artificial organelles. Acc Chem Res 2011, 44, 1039-1049.
  3. Vaiserman, A.; Koliada, A.; Zayachkivska, A.; Lushchak, O. Nanodelivery of natural antioxidants: an anti-aging perspective. Front Bioeng Biotechnol. 2020,7, 447
  4. Gao, S.; Holkar, A.; Srivastava, S. Protein-polyelectrolyte complexes and micellar assemblies. Polymers (Basel). 2019, 22, 1097.
  5. Pippa, N.; Gazouli, M.; Pispas, S. Recent advances and future perspectives in polymer-based nanovaccines. Vaccines (Basel). 2021,9, 558.

………………………………………

  1. Elistratova, A.A.; Gubarev, A.S.; Lezov, A.A.; Vlasov, P.S.; Solomatina, AI.; Liao, Y.C.; Chou, P.T.; Tunik, S.P.; Chelushkin, P.S.; Tsvetkov, N.V. Amphiphilic Diblock Copolymers Bearing Poly(Ethylene Glycol) Block: Hydrodynamic Properties in Organic Solvents and Water Micellar Dispersions, Effect of Hydrophobic Block Chemistry on Dispersion Stability and Cytotoxicity. Polymers (Basel). 2022,14, 4361.
  2. Gou, M.; Zheng, X.; Men, K.; Zhang, J.; Zheng, L.; Wang, X.; Luo, F.; Zhao, Y.; Zhao, X.; Wei, Y.; Qian, Z. Poly(epsilon-caprolactone)/poly(ethylene glycol)/poly(epsilon-caprolactone) nanoparticles: preparation, characterization, and application in doxorubicin delivery. J Phys Chem B. 2009, 113, 12928-33.
  3. Gou, M.; Wei, X.; Men, K.; Wang, B.; Luo, F.; Zhao, X.; Wei, Y.; Qian,, Z. PCL/PEG copolymeric nanoparticles: potential nanoplatforms for anticancer agent delivery. Curr Drug Targets 2011, 12, 1131-50.
  4. Dash, T.K.; Konkimalla, V.B.J. Poly-Ñ”-caprolactone based formulations for drug delivery and tissue engineering: A review. Control Release. 2012, 158, 15-33.

Comment

2)Supramolecular structures of copolymers strongly depend on ratio of blocks in macromolecules. I suggest to add a paragraph dedicated to quantitative analysis of composition of macromolecule - to - supramolecular structure relation.

Answer

According to the reviewer’s comment, we added the following paragraph:

  1. Technology of PEO-PCL based nanosystems for drug delivery

As mentioned above, the supramolecular structures of copolymers strongly depend on ratio of blocks in macromolecules and for this reason, there is an interdependence between the macromolecule - to - supramolecular structure. Namely, PEO-PCL linear and core/shell worm-like brushes showed different %encapsulation efficiencies of low-molecular weight APIs, like doxorubicin. The worm-like brushes exhibited 26% drug loading while the spherical micelles around 50% [36]. Additionally, the solubilization of hydrophobic APIs, like curcumin is also dependent on the ratio of the blocks of PEO-PCL. PEO(5000)-PCL(24500) was found the best copolymer that achieved the highest solubilization of curcumin [40]. For polymersomes, the DNA-PCL-PEG-PCL weight ratio strongly affected the physicochemical characteristics of the supramolecular structures and there was a ratio limitation, in order to avoid large aggregates [57].  The PCL ratio in the group of amphiphilic diblocks, PEG-b-CPCLx, with x = 25, 50, 75, and 100 controlled the encapsulation efficiency of APIs and the pH-responsive release in the different compartments of the gastrointestinal tract [66]. The stability and the release profile of hybrid PEO-PCL nanoparticles is also strongly affected by the molar ratio between the components, i.e. lipid:PEO-PCL ratio. Generally, higher ratio of the polymeric guest stabilized the characteristics of the lipid bilayer and delayed the release of hydrophobic APIs [68,71].

Reviewer 2 Report

The review by Pippa and coworkers highlights some of the major advances on drug delivery systems using PEO-PCL block and graft copolymers as nanocarriers. The review not only provides interesting examples but also conveys the advantages of present systems. I believe that the article will provide useful pointers to references of interest for researchers working in both design and synthesis, as well as application of such materials. In my opinion, this review should be published in Materials after minor revisions.

1.    Errors should be corrected. a) In line 46, “ADEMT” should be “ADMET”. b) “in vitro” and “in vivo” should be “in vitro” and “in vivo” (Italic).

2.    It would have been beneficial if the authors had provided more figures, such as those depicting nanoparticle characterization and drug delivery efficiency, in order to improve the understanding of readers.

3.    It would be helpful if the authors could utilize a table for the different drug delivery systems they discuss.

4.    The authors could have provided more insights on the possible advantages or disadvantages that the nanozyme approach offers in comparison to other approaches aiming at similar applications.

5.    In a final conclusion, the authors could expand more on the potential of the PEO-PCL block or graft copolymers, and discuss some of the limitations and challenges remaining in this area of research.

Author Response

Reviewer #2

Comment

Errors should be corrected. a) In line 46, “ADEMT” should be “ADMET”. b) “in vitro” and “in vivo” should be “in vitro” and “in vivo” (Italic).

Answer

We corrected the above errors in the manuscript.

Comment

It would have been beneficial if the authors had provided more figures, such as those depicting nanoparticle characterization and drug delivery efficiency, in order to improve the understanding of readers.

Answer

In Scheme 1 the PEO-PCL block and graft copolymers as nanocarriers for drug delivery applications are summarized.

According to the reviewer’s comment, we added a new scheme (Scheme 2) for the applications of PEO-PCL block and graft copolymers as nanocarriers:

Scheme 2: Applications of PEO-PCL block and graft copolymers as nanocarriers

We also added Table 1 for nanoparticle characterization and drug delivery efficiency.

Table 1: Techniques and methods for characterization of PEO-PCL based nanoparticles and drug delivery efficiency.

Comment

It would be helpful if the authors could utilize a table for the different drug delivery systems they discuss.

Answer

In Schemes 1 and 2, we summarized the different drug delivery systems and the applications of them.

Comment

The authors could have provided more insights on the possible advantages or disadvantages that the nanozyme approach offers in comparison to other approaches aiming at similar applications.

Answer

According to the reviewer’s comment, in line 524 we added: “The simplicity, sensitivity, and speed of the nanoenzyme technique are its benefits, whereas its cost, point-of-care needs for accuracy and reaction time are its drawbacks.”

Comment

In a final conclusion, the authors could expand more on the potential of the PEO-PCL block or graft copolymers and discuss some of the limitations and challenges remaining in this area of research.

Answer

According to the reviewer’s comment, we re-wrote the conclusion section by adding the limitations and the challenges remaining in this area of research:

PEO-PCL block or graft copolymers, belonging to the family of biocompatible and biodegradable copolymer and exhibiting stealth properties, due to the presence of PEO blocks, show great potential in biomedical applications and most interestingly as drug delivery nanocarriers. As it is depicted by the studies described above, the employment of PEO-PCL in nanocarriers, such as micelles, polymeric or hybrid lipid-polymeric nanoparticles, provides more safe and more efficient drug treatments, by improving drug biodistribution and tissue penetration, also increasing the bioavailability and decreasing the toxicity and side effects. As was already indicated, PEO-PCL and consequently PEO-PCL-based nanoparticles offer flexibility in terms of design from a technological standpoint. Initially, these polymers come in a variety of kinds (compositions and architectures).  These nanosystem physicochemical properties are essential to how they behave both in vitro and in vivo. It is also possible to create controlled release qualities and targeting by using various functionalization techniques. Moreover, PEO-PCL can be easily functionalized, allowing targeting ability. On the other hand, there are just a few products/medicines where PEO-PCL copolymers are utilized. All the nanomedicines in development must undergo a thorough examination of the physicochemical and morphological characteristics of the PEO-PCL formulations utilizing specialist procedures. Preclinical research is more expensive because of this need compared to other pharmaceutical formulations (i.e., tablets, suspensions, etc.). At every stage of the design and development of the aforementioned platforms, as well as for knowledge transfer between academic institutions and the pharmaceutical companies, big pharma also needs scientists (chemists and formulation scientists) with training and great experience in the field of synthesis and characterization of PEO-PCL materials and formulations.

In conclusion, we have discussed a number of PEO-PCL drug delivery nanosystem examples from the literature that have emerged in recent years at various preclinical and clinical study stages. For continued development of these systems as nanomedicines, the results and outcomes were in the majority of the stated situations highly positive. However, the synthesis and the particular molecular characteristics of the copolymer can affect the characteristics of the respective nanocarrier, like size, morphology and drug loading efficiency. Thus, a careful design of polymer synthesis and nanocarrier fabrication should be carried out for achieving better performance of the nanocarrier systems.
